# Playability and Player Experience in Digital Games for Elderly: A Systematic Literature Review

**DOI:** 10.3390/s20143958

**Published:** 2020-07-16

**Authors:** Antonio Rienzo, Claudio Cubillos

**Affiliations:** 1Escuela de Ingenieria Biomédica, Universidad de Valparaiso, Valparaíso 2362905, Chile; 2Escuela de Ingenieria Informatica, Pontificia Universidad Catolica de Valparaiso, Valparaíso 2340000, Chile

**Keywords:** playability, player experience, digital games, mobile health applications, elderly

## Abstract

A higher number of people increasingly uses digital games. This growing interest in games, with different objectives, justifies the investigation of some aspects and concepts involved, such as product quality (game), usability, playability, and user or player experience, topics investigated by the multidisciplinary area called Human–Computer Interaction (HCI). Although the majority of users of these games are children and young people, an increasing number of older adults join technology and use different types of digital games. Several studies establish the increase in learning, socialization and exercise promotion, and cognitive and psychomotor skills improvement, all within the context of active and healthy aging. The objective of this work is to carry out a systematic literature review investigating the player experience of the elderly in digital games. The work allowed answering five research questions that were formulated. The evolution and maturity level of the research area are studied together with the research methods used. The factors that motivate adults to play were also analyzed; what are the recommended technical characteristics for games and some tools and metrics with which games are evaluated for older adults? Research gaps were detected in the area; there are not many specific studies on playability and player experience applied to the older adult, nor are there proven tools and metrics to evaluate them. Particular techniques for assessing and designing games focused on older adults are lacking, and quantitative studies that better identify the factors that affect the playability and experience of older adults in digital games.

## 1. Introduction

As a result of research carried out to date, engineers, health professionals and caregivers, and government entities from many countries have been looking for ways to promote the quality of life of older people. In recent years, people with different knowledge in areas such as engineering, information technology, and medicine have been interested in using technology to help the elderly and improve their well-being. The robotic intervention for the rehabilitation of older adults, the age-friendly mobile phones, the sensor technology for the prevention of falls, prostheses based on 3D printers, and intelligent devices for the care of the elderly are some of the advanced technologies adopted for the welfare of the elderly.

Among these technologies, digital games seem to be promising for improving the physical, social, and cognitive well-being of older people [1,2]. “Digital games” are defined as any game played with an electronic device, either online or independently, for example, a computer, a video game console, a mobile device, or interactive television.

Digital games have been promoted to improve the quality of life of the elderly, according to studies by De Schutter et al. 2008 [3], and Jung et al. 2009 [4]. In particular, and according to Pyae [1], researchers and developers have investigated digital games in terms of improving the experiences of older people when doing physical exercise, their social connection with family and friends, and their cognitive abilities.

Some researchers (such as Pyae and Wang) [1,2] have designed digital games for older people with different goals, such as rehabilitation, cognitive training, entertainment, or improving physical balance. For the design of a successful videogame, the relationship between user experience and commitment, through flow theory, is one of the critical and essential points. Csikszentmihalyi explains that the flow is like a state of being completely “in the zone” [5], and this flow is attainable when there is a tight and optimal balance between skill and challenge.

According to the existing literature, most of the research and commercial projects related to the use of digital games with older people come from developed and wealthy countries such as the United States, the United Kingdom, Australia, Singapore, South Korea, and Japan. There has been only a limited amount of relevant research in developing countries. Most studies have addressed the technical aspects [6] without investigating the perceptions of the elderly and the acceptance of the use of digital games for exercise. These psychological factors are so influential that it is worth considering them from the early stages of development by adopting a user-centered design approach that takes into account the characteristics of the specific target population of the technology solution. Based on these premises, it is necessary to investigate the role of perceptions and the acceptance of older adults towards the use of digital games to identify if the service provided can overcome this preliminary barrier that is so important.

This document reports on an exploratory study that evaluates the state of the art of the concepts of playability and the experience of older adults in digital games. It is organized as follows. Section 2 synthesizes the theoretical framework; Section 3 introduces the theme and studies related to digital games in the elderly; Section 4 explains the process of methodological research; and in Section 5, the analysis performed is detailed, and the mapping results are discussed. In Section 6, the conclusions of the work done are raised.

## 2. Theoretical Background

To understand the concepts of playability and player experience, it is necessary to analyze the subject from different points of view. From the perspective of the quality of a software product (digital game) to the gameplay as the quality of the player’s experience, it is passing through the concept of user experience (UX) and player experience (PX).

### 2.1. The Digital Game as a Software Product

Like any software product, the quality of a digital game can be analyzed under specific standards (such as ISO standards) [7]. However, it is necessary to emphasize that a digital game (like a videogame) is not a typical software system. It was not developed to execute or solve a daily task for a specific and functional objective (e.g., banking site, purchases by web, calculation of remunerations, make graphs from an electronic spreadsheet, supplies inventory). A digital game has a specific objective that is to entertain and make the player feel good while using it. However, it is more subjective, because to a large extent, it will depend on the profile of the player.

For evaluating the user experience (in interactive systems), we can base ourselves on the concept of usability, whose aims have to do with productivity, or how the functionality is well understood and used by the user.

### 2.2. Usability

Usability is a term related to a major discipline known as “human–computer interaction” (HCI), and it is the measure of the quality of the experience that a user has when interacting with a product or system (such as an application or digital game), or interactive service. It comes to represent the discipline that studies how to design software (websites, systems, or applications) so that users can interact with them in a relaxed, comfortable, and intuitive way. Moreover, by considering factors such as easy to learn, remember the operation, the frequency of making mistakes, and the subjective satisfaction of the user (everyone can have different attitudes and experiences, and have different objectives and needs) [8]. According to the international standard ISO 9241-11 (1998), “usability refers to the degree to which a product can be used by specific users to achieve specific goals with effectiveness, efficiency, and satisfaction given a specific context of use” [9], the context of use refers to the conditions or variables under which an interactive product is going to be used; that can provide the environment (space and time), the organization of work processes, or the technical characteristics of the system. The ISO 9126 standard also uses the concept of “quality in use”, which implies similarities with ISO 9241-11, and where six quality features are included [8]. From them, usability or ease of use is the set of attributes that refer to the effort for the product or system to be learned, understood and used, and attractive to the user in a particular context.

It is important to consider usability, as the systems that best fit the needs of the users improve the productivity of the actions; systems that are easier to reduce stress. Moreover, it generates a reduction in the costs associated with maintenance and support. It must be considered throughout the development process, from its beginning until the system, product, or service is available to users. The concept of usability applies to all types of software (organizational, industrial, financial, medical, and also to digital games). In general, usability evaluation methods can be classified as inspections, where small groups of expert evaluators participate, and as usability tests, where user groups participate (as co-discovery).

### 2.3. User eXperience (UX)

When talking about User eXperience (or UX), we refer to the perception that people have when they interact with a product or service (which includes the concept of usability). It consists of the set of feelings or emotions that occur in the user when operating with an interactive system (and that not only includes aspects of skills and cognitive processes and their rational behavior). Users regularly interact with products and services, and the level of satisfaction or achievement they obtain will depend on the experience that each user has in those interactions. Consequently, it will be relevant to meet the target audience, listen to it, understand their needs and interests, create products tailored to their needs, and seek maximum satisfaction from said user.

The UX is defined as “perceptions and responses of the person as a result of the use and anticipated use of a product, system, or service” (ISO standard 9241-210) [9]. The user experience is the set of factors and elements that are related to the interaction that a user has with a specific environment or device. It results in a perception (positive or negative) of said service, product, or equipment, either before, during, or after use. This perception will depend on the factors related to the design (such as hardware, accessibility, design and visual interface, and quality of the contents) and aspects related to the feelings, emotions, and reliability of the product. The functional specifications, applicable to all types of software, also facilitate the user experience.

### 2.4. Playability

For the UX, usability objectives include, for example, performing a task efficiently and effectively; eliminating possible errors; the one that is easy to learn, is intuitive; and that it be designed under standards. The external reward is to perform the work that was pursued. However, for a player’s experience, the objectives of the gameplay include entertaining as long as possible, learning and discovering new things; having fun overcoming goals and obstacles; the inner reward is fun. The concept of playability does not apply to all kinds of software, only to a product, system, or service related to games (of any type).

Playability has recently been studied from several points of view. There is not much consensus on its definition, or the elements that characterize it. According to Gutiérrez and González [10], playability is a term used in the design and analysis of applications and video games that describe the quality of a game in terms of its classic components: objectives, rules, mechanics, and dynamics. Moreover, specifically, it refers to all the emotional experiences that a person can feel and develop when interacting with a digital video game application or system.

The term playability is used in the analysis and design of games, which allows you to describe the quality of the game, considering its rules of operation and its design as a game. It can be said that it is “what the player does in the game” [11], and refers to the experiences of a player during the entire interaction with a game system. A complete definition is “the set of properties that describe the experience of the player before a specific game system, whose main objective is to entertain and entertain satisfactorily and credibly when playing alone or accompanied” [10].

Within the lines of research that address the theme of the gameplay, most of the existing works can be grouped into two trends [11]:Analyze and measure the gameplay as a feature present in the elements of a digital game.Analyze the gameplay as usability in digital games, focusing mainly on its evaluation.

### 2.5. Player eXperience (PX)

As previously defined, the gameplay is the set of properties and variables that describe the Player’s eXperience (PX), and it can be characterized by a set of attributes (such as learning, motivation, satisfaction, effectiveness, immersion, emotion, and socialization) [11]. The current research of PX aims to investigate the cognitive, emotional, and social components, which arise from the experience of personal interaction between players and an application or game system. This focuses on similar UX research lines, although the current research line of PX focuses on the part of the user experience and on everything that happens while the player interacts with the application or the game system [12].

The UX specialized literature also includes the experience that occurs when the user (of any age) interacts with the company that produces the product or service under investigation. Some authors recommended that the term UX be restricted to “products, systems, services, and objects with which a person interacts through a user interface” [12]. This may be relevant for the PX as the company’s brand and its customer service exist in the gaming industry in a similar way to other industries. PX research considers the user state, his experiences at the beginning, and when the interaction with the game ends. PX can only be felt and experienced when a player interacts with a game [13,14,15].

In short, playability is the evaluative process directed towards games, while the player experience is directed towards players. Specifically, the playability methods evaluate the games to improve the design, while the player experience methods evaluate the players to improve the games. This separation of terms becomes essential in the game design process. Figure 1 shows the interfaces between the player, the game, and the game designer, where it can be seen that the playability is aimed at evaluating the game design. In contrast, the player experience must be analyzed in the player–game interaction process [13].

## 3. Digital Games for Elderly

Diverse publications and statistics show that the population of older adults in the world is increasing [16,17,18,19]. As life expectancy increases, so does the cohort of older people, which is currently increasing worldwide and is proportionally higher than at any other time [20,21]. This causes a set of new situations with unique sociological, anthropological, and economic characteristics. Among the least solved problems, there is the condition of disability when facing the needs of daily life, and that it is difficult for many people affected by biomechanical, cognitive, or communication impairments. This disability generates biological consequences such as an increase in functional deterioration, frustration and psychological damage, isolation, depression, and the fact of becoming a load factor for people nearby. There are important studies on dependence in older adults [22]. The self-prevalence of many adults is limited by the impossibility of carrying out daily activities: dressing, grooming, moving, feeding, communicating, and other activities. Older people depend absolutely on others and generally of their own family. This amplifies the problem with an increased impact in economic and psychological terms.

Today’s technological improvements (especially ICTs) provide opportunities to improve the quality of life of the elderly in terms of medical care, physical and social well-being, security, and communication [23]. Worldwide, a large number of technological projects have been developed to help older adults have a better quality of life. Among these technologies, digital games are a promising technology that can improve the quality of life of the elderly in terms of physical, social, and cognitive well-being [1,24,25]. Most commercial digital games have not been configured or customized for older adults.

Interesting research is presented by Aung Pyae and collaborators from the Griffith University of Australia, where he analyzes the potential and the impacts of digital games in the promotion of active aging of older adults, in the case of the country of Myanmar [1]. First, they provide a review of the literature on the theories of active aging and digital games for the physical, social, and cognitive well-being of older people. Second, research gaps stand out, followed by research questions. Finally, they propose a research design to investigate the impacts of digital games on the active aging of older people in Myanmar. They present experiences and exploratory studies of the application of digital games, with Wii Sports games (Nintendo, Kioto, Japan), and with the Xbox Kinect (Microsoft, Redmont, USA), for physical activity, for social action, and cognitive activity. According to the results of the research, the common indicators for active aging are physical activities, social connection, and cognitive abilities. They also highlight that digital games promise to improve the physical movements of older people, social connection, and cognitive well-being. Based on these findings, they proposed a research study that uses digital games to promote active aging of the elderly.

According to Wang [26], the first key to the design of digital games for older users is to design and develop a user-centered model, with complete and comprehensive knowledge and understanding of elderly players. The characteristics of the lifestyle and psychology of the older users must be taken into account, considering the features and implications in the game design for these specific users. Besides, it also analyzes the age-related deficiencies that affect older people to play digital games and reviews the motivations of the elderly in the games and their participation in videogames. This coincides with several authors, such as Brox and Konstantinidis [27], who point out that most commercial games are not suitable for people of this age group for various reasons. Many usability research and user-centered design protocols (UCD) have been developed and applied, but none of them focus on the use of older people’s games for physical activity. Moreover, Kniestedt and others [28] designed an online multiplayer game for mobile phones, called Pocket Odyssey, which helps older adults maintain their physical, social, and mental well-being. With nine design requirements, it was developed to provide an attractive game that positively supports emotion, and encourages online social interactions between players. According to the authors, the mobile application also allows them to play with other older people in a player’s physical environment, and integrates activities that exercise physical skills and cognitive stimulation. In their work, they offer a customizable environment and use the user-centered design approach (UCD), which takes into account gaming and entertainment preferences, opinions about technology, and aspects of everyday life that older people consider important.

Several researchers on the gameplay suggest that “the good playability of a game should be a prerequisite for evaluating the experience of the game” [13]. The design of a game should not contain any problems that may interfere with an individual game experience (especially for older adults). These authors mention that one method to evaluate the game is an expert-based (or heuristic) review, which is a technique to identify problems that can affect application users. It can be used progressively and iteratively at any time during the process of developing the application or game system. According to the authors [13], the above methodology has been used successfully to review and test traditional software (such as information systems or websites), but to be applicable in videogame evaluations, specific heuristics are required. There are currently heuristics for the game, and they are available, which allow experts to evaluate various aspects, such as the user interface that presents the game. However, the area of the development of the heuristics of digital games is still very recent. Consequently, it needs more practical research to create a specialized set of heuristics for games, which can be used to analyze and evaluate any game application or system, and with different groups of players (young and adults), in the homes of the players or on mobile devices. There is a need for heuristics focused on older adults in older players and their special characteristics.

We must also consider age-related deterioration. Aging in older adults is associated with several deficiencies, such as the gradual decrease in vision, problems with hearing, decreased motor skills, and an increase in cognitive disabilities [26]. These characteristics make gaming more difficult in adults. Therefore, the interface design of the application or game system for older adults must have special characteristics: be simple and intuitive, with instructions and complete description, to reduce the cognitive processes (and the memory load) of the adults. The psychological characteristics of the elderly [26] differ from those of the young and cause them certain problems with age. Digital applications or games for older users generally offer some advantages: they help improve psycho-emotional states, such as depression, anxiety, and stress, by providing more interaction, social connections, and more entertainment with other people.

Although older users play digital games less frequently than other age groups, Wang [26] argues that, in general, this cannot be attributed to a lack of motivation, interest, or social interaction. He indicates that although it is a challenge for older adults to interact with new digital technology, studies conclude that they are quite receptive to new devices and computer technology. An investigation into the behavior of technology adoption by high-level players states that older adults would like to spend time in digital applications that can provide more benefits [29]. Other authors [30] also suggest that older people support new technologies by providing valuable opportunities. For example, they do not expect technology to replace personal and face-to-face communication. However, they do desire to connect with other adults with similar hobbies and provide them with additional social activities, and especially to help them stay in touch if they are disabled or immobile.

The measurement of PX has been a challenging and key issue among digital game researchers to understand its impacts. There are different conceptual theories about the motivations for the game, or the most relevant components of the PX. Among other investigations, in Rodríguez et al. [31], they present the correlation between the frequency of specific leisure activities and the satisfaction associated with it, in women and men over 60 years. There is also an extensive number of questionnaires available for the player’s experience, although in most cases, the empirical validation of the scales is limited. In [32], Johnson et al. analyzed two of the most used scales, which are the Player Satisfaction Need Experience (PENS) and the Game Experience Questionnaire (GEQ) [33], and made a detailed study including these scales. However, they apply to all types of players and make no distinction for older adults [34]. The PENS [35,36] represents a model that can be applied and evaluated, which allows measuring the imprecise fun factor of a game. It measures competition, autonomy, and relationship. Competition derives from the challenge, and autonomy is related to a sense of will. The relationship includes the presence of “belonging to a group”, and instinctive controls are the degree of intuition of game controls. In the field of PX, there is currently a wide variety of options to evaluate. There are questionnaires specially designed to evaluate a variety of constructions related to PX, which include presence, enjoyment, fun, immersion, challenge, commitment, satisfaction need, frustration, cognitive learning, and flow. All this demonstrates the multifaceted nature of PX [32]. More recently, in [2], Wang et al. described an integrative theory to support the evaluation of PX, where cases of player experience using motivation, needs, and affection were discussed. In [37], the authors used an event log analysis for the user to experience traceability in real-time.

Dimensions such as the enjoyment of interest and the importance of effort are measures related to intrinsic motivation. Besides, according to expert authors [38,39], the list of positive and negative conditions of the questionnaire (PANAS) was specifically designed to measure the positive and negative effects. The types of questionnaires discussed have recently been used in videogame studies to assess the PX of older people. However, there is not much evidence of case studies. Moreover, there are no studies that indicate that the current scales have been tested or are relevant for older adults, and if they are not, whether they can be adapted or adjusted so that they are.

Finally, Nacke [12,13,40] presents a hierarchical model of game usability that contains specific components, while proposing a methodology to study them for evaluation purposes (although it does not mention the differences between young players and older adults) [13]. Other authors, such as Gonzalez et al. [41,42], also propose definitions and a framework for the game, with a focus on the experience of the player user during the operation of the application or game system. They argue that this approach does not focus on the quality of the game, but on the player, focusing on the player’s experience. This is emphasized in the “Gameplay Model” that they propose (as an extension of the ISO 9241-11 standard), and that contains characteristics (attributes and properties) that define the player’s experience. They also explain that each attribute contains several additional properties. The same researchers also present a model called “Gameplay Facets”, where they explain that different facets allow the identification and characterization of the skill of the game, which is affected by the process of interaction between the application or game system and the players [42,43].

It is estimated that the research area related to playability and player experience is not yet mature. Moreover, there are few contributions to its applicability in older adults. Therefore, it is necessary to carry out a systematic study on the relationship between the gameplay and the experience of the elderly adult user and to deepen the state of empirical knowledge and its recent research on the subject.

## 4. Research Methodology

Systematic reviews of the literature (SLR) have been carried out in a similar way to systematic mapping (SM) [44,45]. These approaches require the systematic application of a protocol and have been applied to a wide range of knowledge domains, and also to identify gaps in a specific domain or area of knowledge.

A preliminary exploratory study was carried out, detecting possible research questions related to the subject and the objectives pursued in the mapping. Then, the best resources and search sites for the documentation to be collected were defined and known. Moreover, with the determined keywords, we proceeded to carry out a systematic process of searching for documents, to later analyze them. The PRISMA (Preferred Reporting Items for Systematic reviews and Meta-Analyzes) statement was used for the process of selecting the information found, which has four stages: identification, screening, eligibility, and inclusion [44,45]; an extraction and synthesis of the results of the studies found were made; and finally, some statistics were made with the information.

### 4.1. Research Questions

In order to guide the literature review, the following questions have been proposed.

RQ1:How has research on playability and player experience evolved in digital gaming applications for the elderly?RQ2:Which are the research methods mostly used in playability and player experience research focused on older adults?RQ3:Which personal factors motivate or condition older adults to play digital games?RQ4:Which guidelines or recommendations exist related to playability and player experience in games intended for older adults?RQ5:Which are the tools and metrics used to assess the playability and player experience of older adults?

### 4.2. Classification of Primary Sources

For the search of primary studies (articles), we proceeded to search the main electronic databases. The keywords that were used were combinations of the concepts of “playability” and “player experience” with the terms “elderly”, “older adults”, and “older people”. The distribution of the results is shown in Table 1. In the “Others” category are included articles that were selected from the references of other articles, and from some systematic reviews (sites such as Research Gate, SAGE Public, Taylor & Francis, Academia.edu, among others). The first search (automatic) yielded a result of 4249 articles.

Due to the high number of publications (although a large percentage of articles are repeated, that is, they appear in several editorials), for a first selection and review of the most relevant articles, it was decided to consider the following criteria.

∗Inclusion: research papers whose titles and abstracts (abstract) included terms that aim to answer any of the questions; published in publishers with a website (and to which you could have access); corresponding to conferences (Proceedings), book chapters, and articles in medium and high impact journals; and in principle, from the year 2010 onwards (although some articles with an earlier date were included), but which were highly relevant concerning the objective of the study.∗Exclusion: articles whose titles and summaries were not related to the object of study, nor the research questions; that was repeated, or focused on the field of digital games with other types of users, and not directly related to older adults, in the areas of gameplay and/or experience as a player.

### 4.3. Analyzing de Search Process

Figure 2 shows the distribution of the 132 documents found in the first preselection by Search Portal (Database). Then (manually) all the preselected ones were reviewed, which yielded a list of 75 articles (of the total of 132 preselected minus the 57 repeated ones), as established in the lower part of Table 1. These selected articles (75), whose complete detail is included in the Appendix A, were able to establish some statistics, such as those shown below. Figure 3 shows the evolution of the number of publications related to the subject in the last ten years (from 2009 up to date).

Table 2 shows the documents found and classified by type: articles, conferences (proceedings), or others (thesis or book chapters). Their percentages are plotted in Figure 4. Figure 5 illustrates the proportion according to whether the document responded to a systematic review or mapping, to a study and/or proposal, or a experimental case study, according to Table 3.

The main specialized authors out of the 75 documents (with the largest number of publications on the subject) are listed below.

José Luis Gonzalez: Professor in the Department of Software Engineering, University of Granada, Spain. He works on most of his articles with his colleague Fco. Gutierrez.Aung Pyae: Doctor in Computer Science, Department of Future Technologies, University of Applied Sciences, Turku, Finland. He works on most of his articles with colleagues M. Luimula and J. Smed.Lennart Nacke: Doctor of Computer Science. He has worked in several Universities and is currently Director of the group “HCI Games and UX”, of the University of Waterloo, Ontario, Canada.

Of these 75 articles of the 1st selection, we wanted to know more about the researches and contributions of the researchers, specifically to the concept and applicability of the gameplay in the elderly and on the experience of these as players of digital games. It was decided to make a second selection, with the main articles that allowed to answer the five research questions raised at point 4.1. Now, based on the detailed contents of the keywords, the introduction, and the conclusions, and prioritizing the experience cases of the older adult player, the analysis yielded a final total of 34 articles.

Figure 6 shows the PRISMA scheme of the document selection process where 34 articles were finally selected, which are attached in Appendix A (with orange color) in the Appendix A. The PRISMA statement is a publication guide for research, and has been conceived as a tool to help improve clarity and transparency in the publication of systematic mapping and reviews (and meta-analysis), and applies to all types of reviews.

## 5. Analysis and Discussion

To analyze the trend of the selected articles, regarding the concepts of “playability” and “player experience” focused on the elderly, it is convenient to define what is meant by older adult in the context of Human–Computer Interaction (HCI) investigations.

It is key in a good HCI investigation to “know your user”, so the question of who the elders are is important. In general, within the HCI and other closely related disciplines (such as psychology), older people are defined as people 60 years of age or more. However, there is no consensus on its characteristics; since researchers have recognized that older people are a very heterogeneous set of users in regards to their experiences, skills, health status, and abilities. Moreover, other previous studies have suggested that a high amount of HCI research presents older adults as a homogeneous category, that is, a set of users with common needs, abilities, and interests [46].

Both visions are different. First, homogeneity proposes us to classify the group of older people according to a set of common characteristics. However, this similarity does not consider the great difference between their life experiences, and they will determine their motivations and interests to use the new ICT tools. Second, the heterogeneous vision, which delivers a more realistic picture of older people, assumes that these people would have a categorization that is easier to define. For example, some studies specified adulthood as the relevant criterion for characterization, although others focused on personal skills or the different transitions experienced in the stages of their lives [46,47].

Moreover, if we move from the HCI context to disciplines such as sociology, the definition of the category of older persons or older adults may be more complex to specify. Some specialists in gerontology (both social and cultural) argue that old age can be considered a social construction that can be defined through everyday discourses and practices. Moreover, researchers in [46,48] have shown that designers and developers generally create their representative models of older adults. Possibly these representations do not always correspond to the identity and personal characteristics of its participants, which can diminish and frustrate the acceptance of the technology.

For this reason, it is considered important that the analysis and results of this work be considered for the design and development of digital games for older adults. Five research questions were asked when facing this research.

### 5.1. RQ1: How Has Research on Playability and Player Experience Evolved in Digital Gaming Applications for the Elderly?

Concerning the first research question (RQ1), it can be affirmed that there is a progressive increase (on average) of publications about issues related to usability, user experience (UX), playability, and player experience (PX), in global terms [13]. Although in the segment of older adults the number of recent publications is lower, the trend shown in Table 1 and Figure 2 is clear.

Starting in 2014, there is an increase and a constant in the number of specific publications on the subject. It is highlighted that most of the documents are conference proceedings (58%) and that 62% corresponds to case studies with older adults.

According to the majority of the publications analyzed, the trend is to focus on how digital games can influence the active aging of older people. The common indicators are physical activity and exercises, followed by their participation or social inclusion in the community, and finally, for the improvement of their cognitive abilities [48,49,50].

According to Ijsselsteijn et al. [48] and Nap et al. [51], most of the digital games that are currently on the market are aimed at a younger audience and contain content that generally does not resonate well with older people. They also state that although the percentage of “older” players is constantly increasing, little is known about them. It is not clear what kind of games older adults play, what motivates them to participate in digital games, what their game needs are, what problems and difficulties they have concerning digital game interfaces, and what their perceptions and experiences with digital games are [52]. Consequently, it is necessary to have a deeper knowledge about older players, so that future applications and digital game systems can be designed with a focus on the end user, so that they are accessible, useful, stimulating, and fun for those specific players.

Some specialists suggest in their work (such as Sayago et al., 2016) [47] that most of the literature that has been reviewed demonstrates that a great advance of ICT projects is being developed. Still, there is a minimum amount of research evidence on the effectiveness of these technological advances.

Complementing Figure 3, when analyzing the evolution of the publications by the previous classification (three focus groups), the data show some significant trends. Of the playability jobs, 44% are articles from 2015 onwards (56% are backward), and for player experience works, 43% are articles published from 2015 onwards (57% are from previous years). The greatest trend is seen in general publications on digital games in adults, as 19 are from the year 2015 (53%) and 17 from previous years (47%).

The increase in interest in investigating these topics is seen in the fact that for the case studies, 22 (48%) correspond to publications from 2015 onwards, while 24 (52%) are from previous years. In the case of proposals with little or no empirical data, ten publications (40%) are from 2015 onwards, while 15 (60%) correspond to previous years; all systematic literature reviews are from 2015 onwards. Although the number of publications related to the topic has increased, several of them cover aspects of usability and user experience. Still, few analyze the gameplay and experience of the older adult player and do not have the appropriate tools or metrics.

Finally, the 75 documents analyzed in the first selection are classified as based on case studies (Appendix A), proposals without empirical data (Appendix A), and reviews (Appendix A). Additionally, a classification was made according to the contribution focus; if the contribution was to the field of playability, player experience, or digital games use by the elderly (Appendix A). The Tables are included in the Appendix A.

According to the analyzed authors, if older adults are expected to benefit from digital gaming applications, it seems necessary to do more research about playability and the player experience, to involve more users, especially those who do not frequently participate in games, to investigate possible barriers and difficulties, or the reasons they may have for not using digital games or applications, which could reduce and resolve this rejection through new types of applications or games.

From the analysis of the publications, it could be verified that in recent years there has been an increase in publications that study the application of digital games in older adults. However, they mainly analyze usability and user experience cases but do not delve into the concepts of playability and player experience. The methodology used for gameplay studies is mainly based on models and proposals without empirical data, while the research on the player experience is based on case studies. It is necessary to delve into different aspects focused on the use of digital games for the elderly.

### 5.2. RQ2: Which are the Research Methods Mostly Used in Playability and Player Experience Research Focused on Older Adults?

To properly answer the second question (RQ2), a first classification was to determine the focus of the study in three groups. According to Appendix A, there are 16 articles on the topic of playability, 28 on the player experience, and 36 cover the topic of adult digital games in general (without delving into the concepts of playability and PX). It is highlighted that five articles were found in which their content includes the subject of playability and PX together [13,41,49,52,53]. A quantitative analysis shows that for playability-related research, proposal studies without empirical data are preferred, while for player experience studies, more case studies are used. Moreover, in Appendix A, it is verified that no systematic reviews were found on the playability and player experience problems; these only analyze general topics related to digital games for older adults, which shows that it is still a developing area of research.

#### 5.2.1. Playability

Making a more detailed analysis of the publications related to “playability”, a large number of them state that usability is no longer sufficient to measure player satisfaction and that it should be expanded with other attributes and properties that describe the experience of the player when participating in a game [10,42,43,54,55]. The most used methods in case studies are interviews and questionnaires. For example, in [56], two cases of playability evaluation in social games are presented, through the use of gameplay heuristics (in the first with 18 evaluators, and in the second with 58 evaluators), reaching six heuristics for specific domains. A similar case was in [57] where the authors defined a set of 43 heuristics for four domains (gameplay, game history, mechanics, and usability), and where they were evaluated by four users with gaming experience, using satisfaction questionnaires. In another case study [49], a game for physical exercises was evaluated, through a usability evaluation, with 21 older people, through observations of the participants. In [53], the gameplay and user experience are analyzed through a video game and where 35 people participated. They were given a pre-test based on questionnaires during the test they played, and finally, they were post-tested through interviews and questionnaires to learn about their experience as a player. The authors of [28], based on user-centered design for a mobile game, conduct an exploratory survey and a series of workshops with a group of 17 users. In [55], the authors, based on a proposed model they called PQM (Playability Quality Model), evaluated the playability of a sports game with 18 people, who took a post-intervention questionnaire to evaluate certain factors of the model (such as satisfaction, effectiveness, and efficiency).

As mentioned, most of the publications about playability are of the proposed type (without empirical data). A large number of these works present models with metrics to evaluate certain parameters of the playability (such as satisfaction, learning, motivation, and socialization), or design proposals and recommendations to achieve satisfactory gameplay of the game by the participants. It should be noted that a large number corresponds to the line of study of the Spanish researchers González, Gutierrez, and Padilla [10,41,42,43,53,55].

#### 5.2.2. Player Experience (PX)

Analyzing in greater depth the works related to the player’s experience, most of these correspond to case studies, where the most used methods are semistructured interviews and questionnaires, and where the CEGEQ (Core Elements of the Gaming Experience Questionnaire) and SUS (System Usability Scale) scales are used to evaluate the results. For example, a group of specialists [14] carried out an experimental design, where they evaluated the player experience through three-game configurations with increased levels of social presence. Forty adults participated, who evaluated a game in pairs, in cabins specially enabled for the experience. They measured the PX through different GEQ subscales. In [58], an experimental intervention with cognitive games is carried out, comparing the experience of the player with two groups (86 youth and 39 adults) with different levels of experience in games. Using post-game surveys, it was verified that the adults did not make more mistakes than the youth, but they took longer to complete their tasks.

The authors of [59] analyzed the effects of different levels of visual complexity on performance and PX of the adult player. With pre-test and post-test questionnaires and semistructured interviews, they evaluated the results. In another work [60], they measured usability and analyzed the PX using a game based on the Kinect device. Through questionnaires and observations of psychotherapists, they achieved 86.3% satisfaction of the 14 participating adults with limited experience. They obtained a series of statistical data using CEGEQ and SUS. In a similar case [13], they evaluated PX through studying the usability of an interactive game of physical activity, with 10 people divided into two groups. There were three sessions, where they were given a tutorial on the game, then they participated in the game, and in the end, they were asked a questionnaire for each session. At the end of the intervention, they were given a post-game interview.

Moreover, in [61], the preferences, usability of the application, and the PX of 14 older adults were evaluated through three sets of physical exercise. They carried out the experience in a specially equipped room, and in the end, a semistructured interview was applied and where the results were evaluated with the SUS scale. Another case study [15] consisted of analyzing the preferences and the experience of the player, and how this varies with age. Two-thousand-seven-hundred-and-forty-seven players (youth and adults) participated, questionnaires that were evaluated with the PENS and IMI scales were used, and a meta-analysis of 11 different related studies is performed.

As seen in the literature review, in the case studies on the player experience, the most used methods are semistructured interviews and questionnaires, with the scales like GEQ (game experience questionnaire) and SUS (system usability scale) to evaluate the results. Still, none of them is specific or has been adapted to be applied with older adults. A more specific evaluation methodology is missing for them.

### 5.3. RQ3: Which Personal Factors Motivate or Condition Older Adults to Play Digital Games?

Regarding the third research question (RQ3), there are several approaches to the personal factors that can motivate older adults to play digital games. The selected articles state that “little is known about the preferences and motivations of older adults in the game” [62], and that the experiences of the older adult player reveal a wide variety of reasons to play. In addition, accumulated knowledge does not contain a good theoretical basis to explain many of the differences observed among older people who play with applications or game systems [47]. According to the authors analyzed, involving users in the analysis and design process to create ICT applications is a fundamental principle of HCI, so that they are more meaningful and easier to use. However, older people generally have not participated in the design and development stages of the majority of the applications, and digital games focused on the elderly as the main target.

Specialized researchers, such as De Kort [14], Nap [51], and Pyae [63], have studied and paid considerable attention to the preferences, motivations, and practices that are recorded in the gaming experiences of the elderly. This research is based on instruments such as interviews, surveys (of different types), and observational studies and has been carried out with older active players or older people interested in playing digital games. According to the authors, preferences for playing certain types of games have also been analyzed (games with simple rules that can be played satisfactorily in a short period, and that involve some cognitive challenge). There are several motivations that adults have for playing digital games, and those are revealed in the results, for example, to connect with your children or grandchildren and family members or relax with former friends.

The question of whether the games developed so far are those that motivate the elderly, and who want to play in their free time and their daily lives raises as crucial. The objectives of game applications or systems for the elderly (such as physical activities, intergenerational games, health applications, or rehabilitation games) do not seem to coincide with the results of studies on the player’s experience. Older people tend to show interest and preferences for games that stimulate their cognitive abilities, or that are intellectual challenges. Moreover, although the evidence of interventions to improve cognitive abilities is often somewhat controversial or inconclusive [47], the potential of digital games to help improve the quality of life and social interactions between older adults, is also confirmed in some research [50,51,64]. However, the evidence of these investigations is also a bit weak, as it is often based on special evaluation studies (conducted in short sessions and with small samples of older people).

As mentioned earlier, older people prefer to play games that present intellectual challenges, and sometimes compare the types of old non-digital games they met or played in their childhood and youth. According to the work of Sayago et al. [47], due to their varied motivations, the game practices of the elderly can be quite different. For example, play learning games on your personal computers or tablets, or participate indirectly while watching your children and grandchildren play video games with their friends. In summary, there is a wide range of personal motivations that older adults have to participate more and more in activities with digital games, such as staying mentally active, having greater social contact, staying busy and having fun, and also to continue acquiring certain knowledge.

In another study by Wang [26], it is suggested that it is necessary to consider the deteriorations related to the age and health of the elderly. People change with age and, in doing so, psychosocial and functional changes occur that affect vision, hearing, movement, cognition, and their relationship with themselves and with others. He mentions that aging is often accompanied by physical or cognitive restrictions, which affect the predisposition of older adults to play.

The above problems have led researchers to study a series of aspects related to usability and user interface for the special population of older adults. In Silva et al. [65], through a list of heuristics, they research for an alternative to interfaces, more traditional and usual computer interaction methods for older adults, and the adoption of the Internet through mobile devices. They conduct more comprehensive studies that investigate the general use of technology and explain how to design screens and interfaces for applications and gaming systems for older adults. In addition, some devices, such as those with touch screens, allow more direct interaction with applications, offering a new range of possibilities for this type of person. In Vasconcelos and Silva´s study [66], they designed and evaluated a gaming platform based on the use of tablets for cognitive training. The results showed that older people were motivated, easily interacted with, and willing to use the platform, as such devices generally required less training, and are more suitable for novice users or those who do not wish to memorize commands or instructions, and at the same time, it incorporates entertainment and socialization as motivational mechanisms. There are also studies on educational augmented reality mobile games [67].

From the analysis carried out, the main factors that motivate older adults to play are socializing, spending leisure time, staying physically and mentally active, interacting, and entertaining themselves with grandchildren and family members. To better understand their motivation and preferences, it is necessary to better profile older adults, applying instruments of motivation psychology, such as IMI [68], satisfaction/frustration scales [69], aspirations index [70], and basic psychological needs scales. With this, you can better understand the target group and with it develop specific techniques for the design and evaluation of games for them. Moreover, on the other hand, with that profiling, you can develop smart games that dynamically adapt to the needs/motivations of an older adult player.

### 5.4. RQ4: Which Guidelines or Recommendations Exist Related to Playability and Player Experience in Games Intended for Older Adults?

In relation to the fourth research question (RQ4), it is still difficult to characterize the playability and experience of the older adult player. Some authors present a framework that attempts to combine empirical and theoretical investigation of player experience (PX) into a unique framework that encompasses the different levels or views of PX [12]. While the full understanding of what constitutes the player’s experience during periods of interaction between users and games has not yet been achieved, there is general agreement that it is a central component of the design and development of user-oriented games, formed by subcomponents of psychological experience, some of which can be measured using psychophysiological methods, above all, in the case of older adults.

According to IJsselsteijn et al. [48], the degree of usability and accessibility of a digital game indicates the ease of use that an older person has to access any content, regardless of their physical, educational, social, psychological, or cultural conditions. They mention that it is essential that websites and digital games are properly designed so that people with some disabilities (such as older adults) can also use them (such as spatial memory, working memory, motor capacity, or visual functions). Due to the rise of ubiquitous societies and their dependence on mobile technologies, smartphone applications are expected to play an increasingly important role in promoting health and wellness. This development is becoming important to the growing population of older adults.

Other analyzed articles make a series of recommendations for the design and development of digital games for the elderly. Certain diseases of old age and the limitations mentioned above should be considered in the design of a digital game. According to the analyzed literature, older adults applications need to have different characteristics from those developed for younger people [71], among them, that older people do not like games with a violent context, that there is the possibility of choosing different levels of play, and that they do not consider a time limit. In addition, some studies provide evidence of the links between personality traits and the gaming experience [12], which gives information on how individuals experience video games, and possibly informs the identification of subgroups of players who are more likely to experience positive or negative impacts when playing video games. An interesting doctoral thesis by Gonzalez [11] analyzed the common problems and potential reasons that may decrease the satisfaction of older users before a game: the impossibility of knowing how to play a game well, completing a task or puzzle, following the historical argument, or of using certain hardware. Moreover, in [41], a detailed characterization of the player’s experience in video games were developed, based on seven attributes (satisfaction, learning, effectiveness, motivation, immersion, emotion, and socialization), each with its properties.

A group of researchers [66], when designing and evaluating a platform for games aimed at older people, raised a set of rules and recommendations. They recommend the use of mobile devices since they can be transported and played anywhere, allowing older people with some limitations to play more easily, in a more comfortable position and environment. They also state that, to maintain the user’s focus and attention on the objective of the game, the user must have clear objectives and goals, and that they act as challenges to keep the adult involved in the game. Moreover, providing more than one game or alternative allows these people to choose their games by swapping between them, preventing them from getting tired of a single game. These characteristics are relevant and favor the gameplay and the experience of the player.

From the review of selected articles, there are few studies focused on gameplay for older adults, except for previous work by Wang [26], Rodriguez [31], and Marston [62]. However, these authors do not refer specifically to the attributes of the game or the properties that characterize it. The contribution of Paavilainen [40], which in his articles proposes a new definition of “game-centered game”, is highlighted. Its purpose is to define the game in a generalized way that is possible to apply to all types of games, whether digital or non-digital (physical). His approach does not include external factors such as user experience or player sociability, which are subjective or depend on the context in which the player performs. In his work [40], the basis of its definition is that there are at least three components that define playing a game: functionality, usability, and playability. The first is defined by technical quality (how the game works), usability focuses on the user interface quality (elderly difficulties), and playability is concerned with the quality of the rules of the game (interaction of what happens with the player). If any of these three components is defective, the game does not work as it should, the game is negatively affected, and it cannot be easily used or provides unexpected or meaningless results. However, it also does not establish differences in games for older adults.

Consequently, when an older adult uses a digital game, it should have been designed or adapted (or taken into account) to the health conditions or limitations of age. In this way, interaction with the system will be more rewarding. Digital games, to have a good degree of gameplay and offer a satisfactory player experience, must consider aspects such as an appropriate interface (design, format and adequate visual representation); that offer customizable fonts or icons, and should be easy enough to use to allow a wide range of users. Older people favor games that portray real-world themes, and by customizing a game to the person’s real world, it is possible to provide a better gaming experience (using familiar images, videos, or sounds). In Teixeira et al. [71], several similar elements are also raised that exert a positive and negative influence on the motivation of older people for digital games.

In summary, comprehensively modeling how the game is created and experienced and the gaming experience with older adults is currently not straightforward; the large number of variables that can affect them is too large to take them to any type of functional model even if it could exist. There is little involvement of older adults in game design. This therefore remains a matter of further research in the area.

### 5.5. RQ5: Which Are the Tools and Metrics Used to Assess the Playability and Player Experience of Older Adults?

The fifth question (RQ5) was the most complex to answer due to its diversity. The research allowed us to find a complete list of heuristics to analyze and evaluate smartphone applications aimed at the elderly. In the work of Silva et al. [65], these were grouped by developers and programmers according to the following categories; perception (such as limitations that occur with age, such as hearing loss or changes in visual acuity), cognition (such as handling a large number of elements through operational memory or maintaining attention), dexterity (such as difficulties in motor skills), navigation (as the user understand the structures of the applications and move through them), content (related to the language and type of information used in the applications), and visual design (such as design details, format, and representation of visual content). This is considered an interesting contribution, as the results showed that all the heuristics in the list were useful in evaluating applications while conducting a heuristic evaluation. However, the authors [65] considered that the research could be supplemented by usability testing with older adults as end users, as no expert inspection method supersedes user testing.

The reviewed literature suggests that there are still challenges to overcome: measuring experimental dimensions and evaluating the degree of motivation, fun, and emotion is more difficult to achieve, compared to measuring traditional performance metrics, such as the time spent playing, the level of the game reached or the number of tasks completed at each stage of the game. In Ibrahim et al. [72], they stated that “it is difficult to obtain knowledge about what the players did when playing and knowing how different game design elements affected their experience of interaction with the game”. As mentioned, in the case studies, the most widely used methods are semistructured interviews and questionnaires, with the CEGEQ (Basic elements of the gaming experience questionnaire) and SUS (System usability scale) scales to evaluate the results; however, none of them are specific or have been adapted to be applied with older adults.

In Ibrahim et al. [72], they also propose to measure the experience of the player, with the use of attributes and properties of the game that allow us to measure whether a player has fun or not when playing a digital game [13]. The characteristics of the gameplay characterize PX by analyzing and evaluating all the variables and aspects of the behavior, and the feelings and emotions of the player when interacting with the game. According to the authors in [72], an adequate value of the game allows the player to have a more positive gaming experience, which results in a greater willingness to assimilate the characteristics of the game. These authors also state that the aspects considered in the PX represent the characteristics of the interaction process (such as sensation, motivation, feelings, evaluation, and commitment), as well as user satisfaction, and also the experience acquired (while the activity time is carried out in the application or game system). However, concerning the game, they make no difference in terms of user ages (young players, adults, and older adults).

Researchers from the University of Granada [73] propose a model (PQM) to guide the evaluation process of the user’s experience. They establish a set of attributes and properties to characterize and measure the quality of the game and the experience of the player. This PQM model (quality model of the experience of use based on the Gameplay) presents five factors and attributes of quality (effectiveness, efficacy, context coverage, freedom of risk, and satisfaction). They also propose a set of metrics based on the gameplay of a videogame [73]. However, again, without making a distinction by player’s age.

Therefore, although there are some model proposals, certain tools (such as some heuristics) and metrics to assess the degree of player playability and experience, are not specifically designed or adapted for older adults. Here an important gap is detected: there are some models proposed but few specific case studies. No methodologies were found that allows designing and evaluating games aimed at the elderly or that consider their relevant characteristics.

### 5.6. Advantages and Gameplay

Several specialized studies also show the advantages of playing during adulthood, even during aging [50]. As Ijsselsteijn [48] and Pyae [49] state, although additional studies are needed, actual research has indicated that digital games can have a positive impact on the quality of life of the elderly. For example, and according to Teixeira et al. [71], some hormones are released when there is fun and fun activity. As a result, games can be useful for provoking mental stimulation, improving well-being and increasing self-esteem when a player defeats an opponent, overcomes a challenge or wins a prize. Additionally, these same authors mention that games can help to improve the physical-motor skills of the elderly. Even competitive and action games, which are not liked by older adults (due to the rapidity of movement they require and the intense concentration and interaction), can have positive effects in older adults regarding their health, helping to stop or decrease the natural deterioration of cognitive abilities [48].

According to the analyzed literature, despite all these advantages and benefits, more research is still needed, particularly to study which factors and characteristics motivate this population to start and maintain the habit and interest in playing game applications or systems [74]. Moreover, the evaluation of the games is proposed for a longer period, and with larger samples.

## 6. Conclusions

Carrying out the systematic review confirmed that although several researchers have analyzed the concepts of usability, user experience, and playability (including the player’s experience), it is a relatively new area, so there is still little information (and research) about their applicability in the games for older adults. From the analysis of the publications, it could be verified that in recent years there has been an increase in publications that are studying the application of digital games in older adults. However, they analyze cases of usability and user experience but do not delve into the concepts of playability and player experience. The methodology used for playability studies is mostly based on models and proposals without empirical data, while research on the player experience is based on case studies.

Currently, “players” are not only young people, and digital games are being widely used by people of increasing diversity (different cultures, countries, ages, and gender, among other characteristics). Digital games have the potential to improve the free time and social connection of the elderly, and also provide physical and mental exercises. Several of the reviewed articles agree that the game for the elderly should not only be aimed at the activity itself, but also to connect and socialize people, personal growth, and contribute to the community and society. For them, digital play represents a means to have a pleasant time, challenge cognitive skills, and make sense of the day. Most case studies (from several referenced authors) indicated that the main motivation of older people to play digital games was to have fun and relax.

However, most commercial digital games that are on the market are aimed at younger audiences and are not especially designer or adapted for adults. The publications also mention that while the number of older players is constantly increasing, little is known about them. It is not clear what kind of games older adults play, what motivates them to participate in digital games, what their game needs are, what kind of games they play, what difficulties they face with digital game interfaces, and what their perceptions and attitude changes about digital games. By gathering a deep knowledge about older players, future applications, or video games will be designed to be more useful, attractive, and fun for older people.

The research is still very early, focusing mainly on demonstrating the benefits in the use of video games by older adults and case studies of particular games or platforms. Progress is required in several aspects: identifying the relevant variables and factors (and the relationships between them) that allow generating what types of benefits for older adults when they play; and validate if the existing techniques, methodologies, and tools to measure playability and user experience are suitable for the older adult group, proposing adjustments and adaptations if necessary.

The player’s gameplay and experience, which is difficult to describe and evaluate in young people, is much more difficult to analyze in the case of older adults. Indeed, there is a research gap in the area. The systematic review allowed answering the five research questions that were asked, and also, to know the limited knowledge so far regarding gameplay and the experience of the player in the elderly. The developed work could be subject to deeper evaluation and analysis, serving as a basis for further research in the subject. In the analysis of the publications, no systematic reviews were found about the concepts of playability and player experience, neither for the general public nor for older adults.

Future work considers analyzing and proposing a guide with essential recommendations intended for inclusion in the development of new digital games, to improve the playability and experience of the player, in older adults, and to test specific tools and metrics to evaluate them.

## Figures and Tables

**Figure 1 sensors-20-03958-f001:**
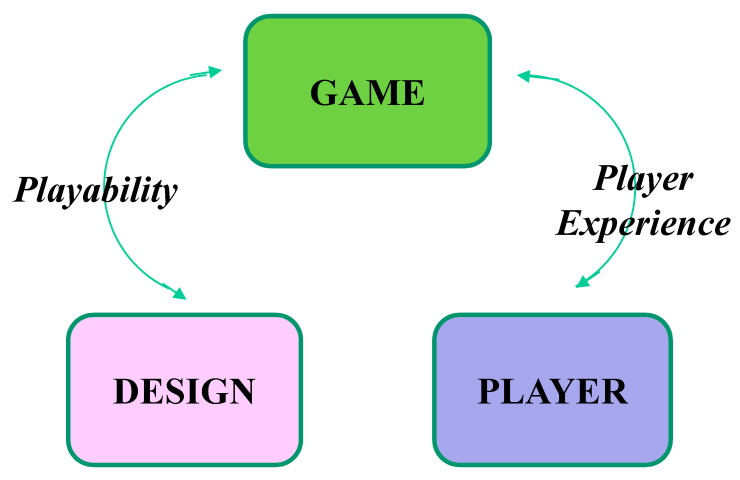
Relationship between playability and player experience.

**Figure 2 sensors-20-03958-f002:**
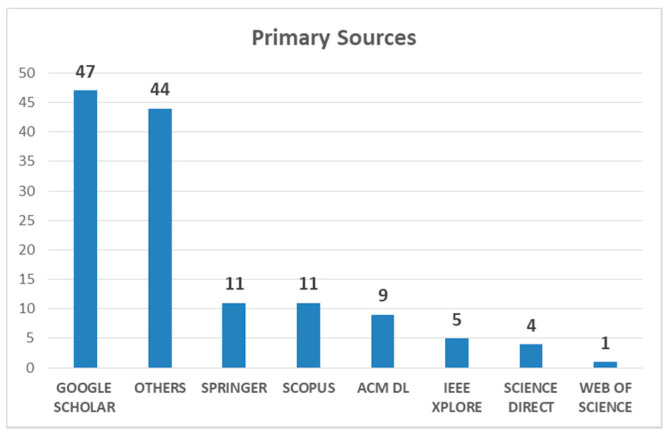
Number of articles found in search sites.

**Figure 3 sensors-20-03958-f003:**
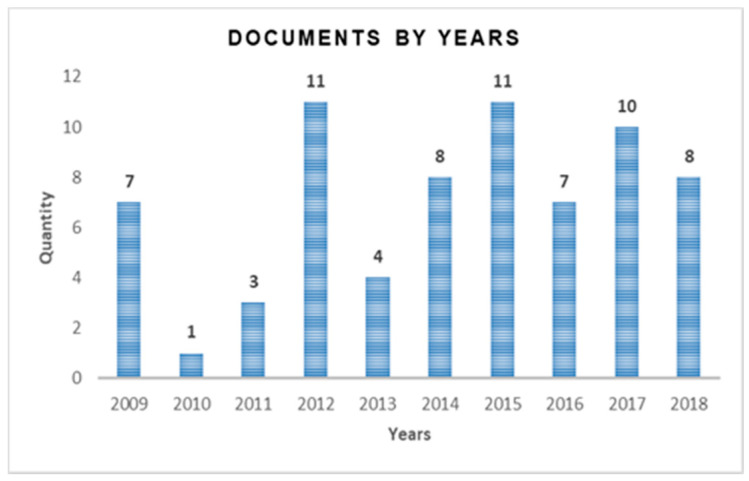
Number of articles found in search sites per year.

**Figure 4 sensors-20-03958-f004:**
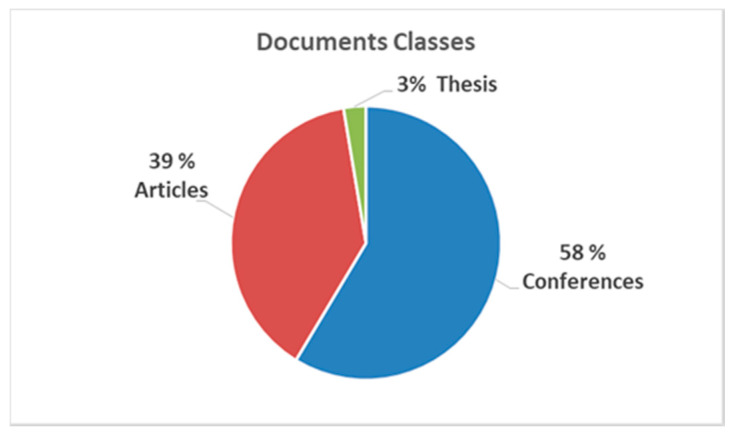
Quantity by classes of documents.

**Figure 5 sensors-20-03958-f005:**
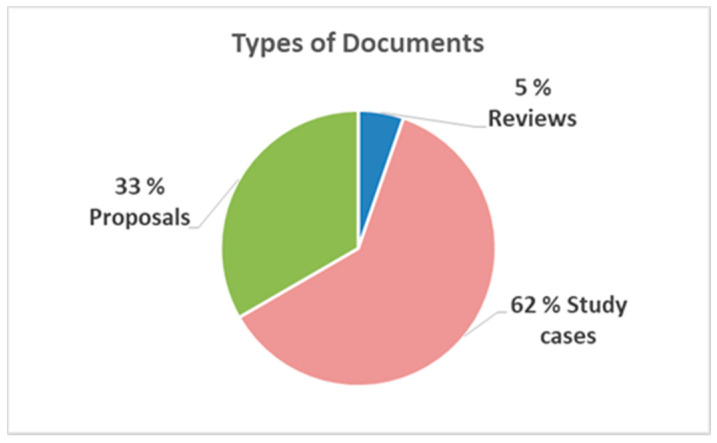
Number of items found by Types of Documents.

**Figure 6 sensors-20-03958-f006:**
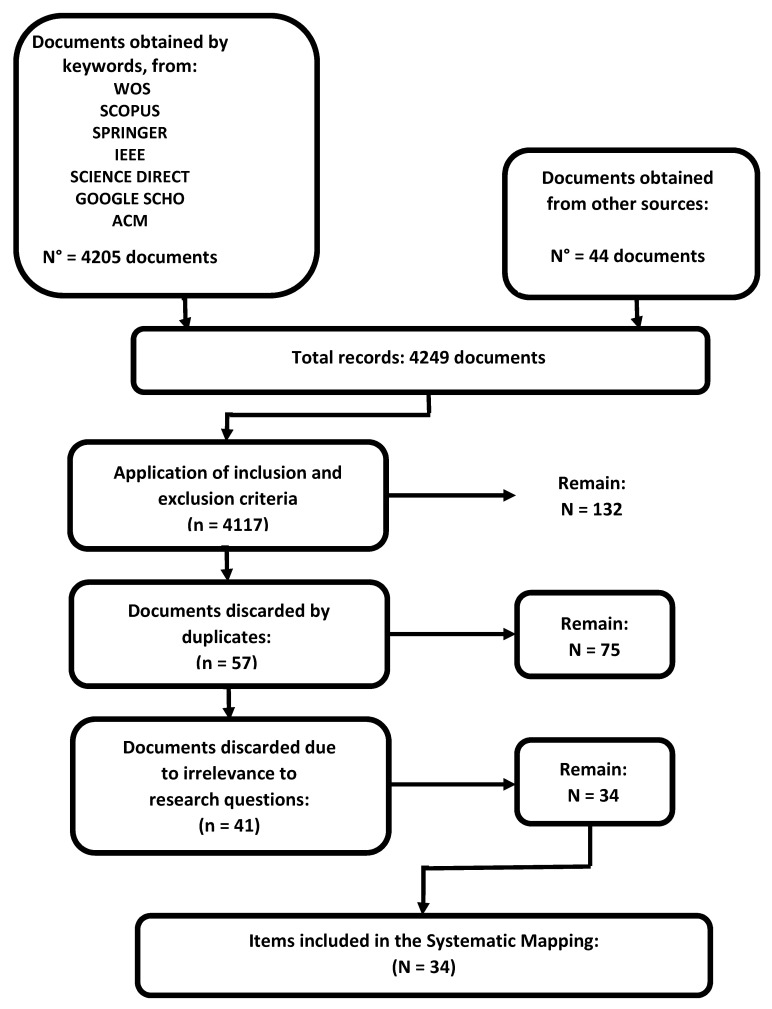
PRISMA scheme for the selections.

**Table 1 sensors-20-03958-t001:** Results of searches in publication database.

Key Words	WoS	Scopus	Springer	IEEE	Science Direct	Google School	ACM DL	Others	TOTAL
Playability & Elderly	1	5	57	1	35	744	0		
Playability & Older Adults	2	1	28	0	19	767	1		
Playability & Older People	2	1	17	0	11	752	0		
Player Experience & Elderly	0	5	87	2	32	716	8		
Player Experience & Older Adults	0	9	51	2	19	549	2		
Player Experience & Older People	0	3	20	1	10	245	0		
Sub Totals	5	24	260	6	126	3773	11	44	4249
Pre-Selected	1	11	11	5	4	47	9	44	132
Less repeated ones									75
1st Selection									57
2nd Selection									34

**Table 2 sensors-20-03958-t002:** Results of searches.

Documents Classes	Documents
Conferences	44
Articles	29
Thesis	2
Total	75

**Table 3 sensors-20-03958-t003:** Results of searches.

Types of Documents	Documents
Reviews	4
Study Cases	46
Proposals	25
Total	75

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
