# Peer review of "Playability and Player Experience in Digital Games for Elderly: A Systematic Literature Review"

_sensors, 2020, doi:10.3390/s20143958_

Round 1

Reviewer 1 Report

The article is well organized and discusses an important subject. Despite that, I was unable to identify a major contribution.

In order to improve the article, I suggest a careful reading to identify English errors. For example:

- In the Abstract, you wrote "Person-Computer Interaction (HCI)", but in Section 5, you described HCI as "Human-Computer Interaction", which is a better known expression.

– When a word has more than one right spelling (e.g. "well-being" and "wellbeing") choose only one of them to use in all the text.

- There are some sentences and titles that should be revised (e.g. "Section 3 introduces digital games in the elderly", "Theorical Background", lines 114 to 116, 150, 215). 

- Avoid repeating words in consecutive sentences. For example, the use of the word "therefore" in lines 421 and 424.

- Correct punctuation errors. For example, in line 457 ("... the elderly can be quite different; For example") and in line 574 ("Researchers from the University of Granada [44], propose").

- Correct the title of section 5.4. "Advantajes".

Author Response

The article is well organized and discusses an important subject. Despite that, I was unable to identify a major contribution.

In order to improve the article, I suggest a careful reading to identify English errors. For example:

- In the Abstract, you wrote "Person-Computer Interaction (HCI)", but in Section 5, you described HCI as "Human-Computer Interaction", which is a better known expression.

– When a word has more than one right spelling (e.g. "well-being" and "wellbeing") choose only one of them to use in all the text.

- There are some sentences and titles that should be revised (e.g. "Section 3 introduces digital games in the elderly", "Theorical Background", lines 114 to 116, 150, 215). 

- Avoid repeating words in consecutive sentences. For example, the use of the word "therefore" in lines 421 and 424.

- Correct punctuation errors. For example, in line 457 ("... the elderly can be quite different; For example") and in line 574 ("Researchers from the University of Granada [44], propose").

- Correct the title of section 5.4. "Advantajes".                                                                                                                                                                      Please see the attachment.

Reviewer 2 Report

The paper surveys recent work on recent work on games for the elderly, focusing on user experience. There are a several areas that the paper should address. 

  1. The discussion of usability, user experience, playability. and player experience, needs to be made clearer. While the paper does include definitions, the distinction between them is not clear. What are the differences? Do papers make use of terms differently? One suggestion is to synthesize the various definitions into the paper's own classification, and then discuss the various publications accordingly. 
  2. The answers to the 3 research questions are not well answered. For example, Section 5.3 answers the question of what aspects are relevant elderly when it comes to digital games. There is no clear answer of what these aspects are. The paper should include a clear answer for each of the posed question.
  3. The survey of the literature in section 5 needs to include sufficient technical details about each of the papers, especially the results of the findings. For example, in Section 5.2, the description of [40] was that they research alternative interfaces like touch screens. What was the result? Key findings? Limitations of [40]? 

Author Response

The paper surveys recent work on recent work on games for the elderly, focusing on user experience. There are a several areas that the paper should address. 

  1. The discussion of usability, user experience, playability. and player experience, needs to be made clearer. While the paper does include definitions, the distinction between them is not clear. What are the differences? Do papers make use of terms differently? One suggestion is to synthesize the various definitions into the paper's own classification, and then discuss the various publications accordingly. 
  2. The answers to the 3 research questions are not well answered. For example, Section 5.3 answers the question of what aspects are relevant elderly when it comes to digital games. There is no clear answer of what these aspects are. The paper should include a clear answer for each of the posed question.
  3. The survey of the literature in section 5 needs to include sufficient technical details about each of the papers, especially the results of the findings. For example, in Section 5.2, the description of [40] was that they research alternative interfaces like touch screens. What was the result? Key findings? Limitations of [40]? 

Round 2

Reviewer 1 Report

The authors added new contributions to the article. As they have adopted the expression "older adults" (or older people) rather than "elderly", I suggest they do the same in the title of the article, sections and research questions. I also suggest to review the sentences that use the word "you", such as "you can better understand the target group", in Section 5.3.